



# Bag-of-words-based anomaly-detection principal component analysis and stochastic optimization for debris flow detection and evacuation planning

Chia-Chun Kuo[1], Yi-Ren Yeh[2], Kuan-wen Chou[3], Chien-Lin Huang[3], Ming-Che Hu[1]

[1]Department of Bioenvironmental Systems Engineering, National Taiwan University, No. 1, Sec. 4, Roosevelt Road, Taipei 10617, Taiwan
[2]Department of Mathematics, National Kaohsiung Normal University, No. 62, Shenjhong Road, Yanchao District, Kaohsiung, Taiwan
[3]hetengtech Company Limited, Rm. 6A, 2F., No. 7, Sec. 3, New Taipei Blvd., Xinzhuang Dist., New Taipei City 242, Taiwan

*Correspondence to*: Ming-Che Hu (mchu@ntu.edu.tw)

**Abstract.** Debris flows are natural disasters, with soil mass, rocks, and water traveling down a mountainside slope. Debris flows are extremely dangerous; their occurrence incurs huge losses to life and property. The purpose of this research is to develop debris flow detection and emergency evacuation systems. A bag-of-words model is established for analyzing the features of debris flow events, and an anomaly-detection principal component analysis (PCA) model is proposed to detect debris flow. Using real-time debris flow prediction and monitoring, a stochastic optimization model for evacuation planning is formulated. Case studies of debris flow detection in Shenmu village and Fengchiu, central Taiwan, are conducted. Shenmu village and Fengchiu are areas of high potential debris flow, and each has a population of around 800 people. The results show that combining bag-of-words and anomaly-detection PCA methods could predict 6 out of 8 occurrences of actual events, providing a prediction rate of 75%. In addition, the models make 13 predictions, and 6 of them are correct, providing a prediction accuracy of 46%. Optimal parameters (including window size, bag length, filter ratio of training data, and anomaly threshold) of the models are also examined to increase the accuracy of debris flow prediction.

Keywords: debris flow, bag-of-words, anomaly-detection, stochastic optimization

## 1 Introduction

Debris flows can be triggered by intense rainfall, floods, earthquakes, snowmelt, dam-breaks, and other geological factors (Promper et al., 2015; Chen et al., 2016; Masaba et al., 2017; Takahashi, 1981; Keefer et al., 1987). Debris flows bring soil masses, water, rocks, and mud down mountainside slopes by gravity. Debris flows are extremely dangerous disasters, and occur frequently on mountainsides worldwide. Serious damage to residents, private property, buildings, roads, and bridges has been caused by debris flows in the past (Wieczorek, 1987; Cheng et al., 2005; Fan et al., 2015; Calvello et al., 2015; Manandhar et al., 2015). Since 2002, the Taiwan Soil and Water Conservation Bureau (Taiwan SWCB) has installed 24 fixed stations for debris flow observation, as well as various monitoring devices in areas at high risk from debris flows. With the data obtained



from these debris flow monitoring systems, this research proposes innovative machine learning methods for capturing debris flow features, and establishes stochastic optimization for emergency evacuations. A smart system for detecting debris flows and preventing slopeland disasters is also established.

This study newly proposes a bag-of-words method to re-interpret debris flow data and enhance computational efficiency. Then, an anomaly-detection principal component analysis (anomaly-detection PCA) model is developed to create a debris flow warning system by detecting data related to abnormal debris flow characteristics. A stochastic programming model is also established for emergency evacuation.

The bag-of-words method is a sequential feature-learning algorithm used for pattern recognition, computational learning, artificial intelligence, simulation, and predication (Lin et al., 2012; Gui and Yeh, 2014). Text documents, sentences, and paragraphs are a conglomerate of words, and the bag-of-words model breaks down text into words and analyzes the frequency of each word.

An anomaly-detection PCA is a statistical method for detecting abnormal data on a principal coordinates system (Candès et al., 2011; Lee et al., 2013). The principal coordinates of eigenvalue problems are identified for different data covariance matrixes. Then, the anomaly-detection PCA detects abnormal data by measuring the impact of new data on principal directions. The impact on principal directions is used to identify abnormal/outlier data.

Stochastic optimization is an optimization model for uncertainty (Barbarosoglu and Arda, 2004; Bretschneider, 2012; Bozorgi-Amiri et al., 2013; Pourrahmani et al., 2015; Wood et al., 2016; Xu et al., 2016; Azam et al., 2017). In the case of two-stage stochastic optimization, two decision stages are separated by uncertain events. The decisions made in the first stage are here-and-now decisions, before the uncertainty occurred; the decisions at the second stage are wait-and-see decisions, after the occurrence of uncertainty. In the case of sequential uncertain events, the model can be extended to multi-stage stochastic optimization. This research applies stochastic optimization for evacuation planning if a debris flow occurs.

Previous studies have applied machine learning algorithms (including support vector machines (SVM), decision tree, logistic regression, Bayesian, fuzzy, neural network, etc.) to analyze and predict natural disaster events (Lee and Pradhan, 2007; Dean and Ghemawat, 2008). Yao et al. (2008) formulated SVM to analyze landslide susceptibility mapping in Hong Kong. They used slope angle, slope aspect, elevation, profile curvature of slope, lithology, vegetation cover, and topographic wetness index (TWI) as environmental parameters to categorize the occurrence of landslides. Marjanovic et al. (2011) compared SVM, decision trees, and logistic regression algorithms. In these machine learning algorithms, geological, morphological, and environmental attributes are used to predict landslide susceptibility assessment. Xu et al. (2012) developed a GIS-based SVM model to examine the earthquake-triggered landslide in the Jianjiang River watershed. Pradhan (2013) compared the decision tree, SVM, and adaptive neuro-fuzzy inference system (ANFIS) for examining potential landslide occurrences in the Penang Hill area. Liang et al. (2012) developed a novel approach for assessing debris flow hazard risk using a Bayesian Network (BN) and domain knowledge. In this study, an SVM and Artificial Neural Network (ANN) were compared with a BN.





Some previous studies have applied bag-of-words and anomaly-detection methods for environmental, biomedical,
human, traffic, and computer science analyses. Khan et al. (2011) established a homography-based visual bag-of-words model
to perform scene localization in indoor environments. Behley et al. (2013) combined multiple softmax regression classifiers
learned from specific bag-of-word representations, proposing a segment-based object detection approach using laser range
data. Wang et al. (2013) applied bag-of-words representation for biomedical time series classification, and their experimental
results demonstrated that the method was insensitive to the parameters (i.e., local segment length and codebook size) of the
bag-of-words model and robust to noise. Baydogan et al. (2013) presented a bag-of-features framework to classify time series
data. The experimental results show that the bag-of-features framework provided better results than competitive methods on
benchmark datasets from the UCR time series database. Gui and Yeh (2014) developed a temporal bag-of-words model for
time series classification. Their method was applied to user identification by a door opening and closing trajectory. Ringberg
et al. (2007) conducted a sensitivity analysis of PCA for traffic anomaly detection and analyzed network-wide traffic
measurements from two IP backbones (Abilene and Geant) across three different traffic aggregations. Brauckhoff et al. (2009)
proposed a spatial PCA for network-wide anomaly detection. They compared the PCA with a Karhunen–Loeve expansion,
since PCAs are very sensitive to calibration settings. Lee et al. (2013) conducted anomaly detection via online oversampling
PCA, and compared with other anomaly-detection algorithms, the online oversampling PCA provided feasibility in terms of
accuracy and efficiency.

Compared with previous, related studies, the significant contributions of this research include innovative development
and the application of bag-of-words, anomaly-detection PCA, stochastic optimization methods, and integrated frameworks for
debris flow detection and evacuation. Case studies in high potential debris flow areas, Shenmu village and Fengchiu, are also
conducted. This paper is organized as follows. Section 2 expresses the methodology, including the bag-of-words models and
stochastic evacuation optimization. Section 3 conducts case studies of debris flows in Taiwan, and the results are discussed.
Section 4 presents the conclusions.

## 2 Methods

This section formulates the bag-of-words model and anomaly-detection PCA for debris flow prediction. With observations
and predictions, a real-time stochastic optimization method is established for debris flow evacuation planning.

The bag-of-words method is introduced as follows (Gui and Yeh, 2014). Eq. (1) defines $X^i$ to be vector variables
representing $i-th$ time series debris flow data. Subsequences with window $w$ are extracted from the time series data. The
number of subsequences of $X^i$ is $p-(w-1)$, since the length of $(x^i_{w-1}, \dots, x^i_1)$ at the end of the time series data is less than
number of windows $w$. Suppose there is $n$ time series data, then the total number of subsequences is $(n) \times [p-(w-1)]$.
Further, in Eq. (2), $y^i$ is denoted as class variables of time series $X^i$.

$$X^i = (x^i_p, x^i_{p-1}, x^i_{p-2}, \dots, x^i_w, x^i_{w-1}, \dots, x^i_1) \qquad \forall i \qquad (1)$$



$y^i = (1, 2, \ldots, c)$ $\forall\, i$ (2)

Time series data are divided into two sets of data for model training and testing. Time series data are disaggregated into subsequences with length w for training. A k-means method is applied to cluster d centroids. The set of centroids are referred to as words for the bag-of-words method. The d subsequences of centroids are the elementary components for predicting unknown time series data.

The test time series data is broken into subsequences, with window w, and the closest centroid is sought for each subsequence. The denote $h_j$ is used as the centroid index for each subsequence. The $h_j = 1$ while jth centroid is the closest centroid, and $h_j = 0$ suggests otherwise. The closest centroid is found for each subsequence, then all subsequences are categorized with vector indices, represented by H in Eq. (3). Only one of $h_j$ is equal to one, and the rest are equal to zero, so the summing of vector indices over all of the subsequences in time series data yields combination of centroid for the data. The

summation vector indices of a time series dataset represent the occurrence frequency of centroids, so the testing time series data can be predicted and categorized.

$H = (h_1, h_2, h_3, \ldots, h_d)$ (3)

An anomaly-detection PCA is applied to analyze the bag-of-words results for detecting debris flow events. PCA is an orthogonal transformation method for dimension reduction [Lee et al., 2013]. The data of observation K is denoted as $data_k$,

and μ is the mean of the observation data. $\sum_k [(data_k - \mu)(data_k - \mu)^T]/n$ calculates data variance and its matrix form represents the covariance matrix V in Eq. (4). If U is the unit vector of principal coordinates, multiplying the left by $U^T$, and the right by U, yields the variance of the principal coordinates data in Eq. (5). PCA then determines the principal coordinates by maximizing variance in Eq. (5). Solving the variance maximizing problem of PCA produces eigenvectors, U, of the covariance matrix, COV. Alternatively, the principal axis can be decided by minimizing the data reconstruction error in Eq. (6).

The data reconstruction error is equal to the original data, $(data_k - \mu)$, minus data reconstruction, $UU^T(data_k - \mu)$.

$$V = \sum_k [(data_k - \mu)(data_k - \mu)^T]/n \tag{4}$$

$$\underset{\|U\|=1}{MAX} \sum_k [U^T(data_k - \mu)(data_k - \mu)^T U] \tag{5}$$

$$\underset{\|U\|=1}{MIN} \sum_k \|(data_k - \mu) - UU^T(data_k - \mu)\| \tag{6}$$

The anomaly-detection PCA measures the impact of abnormal outlier data on principal directions. If A represents all of

120 the observed data, then leaving one data, $x_t$, out of A gives A1, i.e., $A1 = A\backslash\{x_t\}$. $V_A$ and $V_{A1}$ are the covariance matrices of A and A1, calculated in Eqs. (7) and (8). The principal coordinates are determined by solving the eigenvectors of the covariance matrix in Eqs. (9) and (10). The impact of abnormal outlier data is measured by the differences of PCA principal directions in Eq. (11). A larger number of s in Eq. (11) indicates a large change in principal directions by anomaly data.

In contrast, adding one data point $x_t$ to A yields A2, i.e., $A2 = A \cup \{x_t\}$. $V_{A2}$ is the covariance matrix of A2, calculated

in Eq. (12). The updated eigenvector is estimated in Eq. (13). The impact of adding new data is measured by Eq. (14).





Changes in principal direction by adding just one data is not significant if the original size of the dataset is relatively large. Duplicating $x_t$ by $n1$ times and adding $\{x_t, x_t, ..., x_t\}$ to A yields A3, i.e., $A3 = A \cup \{x_t, x_t, ..., x_t\}$. $V_{A3}$ is the covariance matrix of A3, calculated in Eq. (15). A new eigenvector is calculated in Eq. (16), and the impact of adding n1, new data, is measured by Eq. (17).

$$V_A = \frac{1}{(n)}\sum_{x_i \in A}(x_i - \bar{x}_i)(x_i - \bar{x}_i)^T \tag{7}$$

$$V_{A1} = \frac{1}{(n-1)}\sum_{x_i \in A\setminus\{x_t\}}(x_i - \bar{x}_i)(x_i - \bar{x}_i)^T \tag{8}$$

$$V_A(u_A) = \lambda_A(u_A) \tag{9}$$

$$V_{A1}(u_{A1}) = \lambda_{A1}(u_{A1}) \tag{10}$$

$$s = 1 - |\langle u_A, u_{A1}\rangle/(\|u_A\|\|u_{A1}\|)| \tag{11}$$

$$V_{A2} = \frac{1}{(n+1)}[(x_t - \bar{x}_i)(x_t - \bar{x}_i)^T + \sum_{x_i \in A}(x_i - \bar{x}_i)(x_i - \bar{x}_i)^T] \tag{12}$$

$$V_{A2}(u_{A2}) = \lambda_{A2}(u_{A2}) \tag{13}$$

$$s = 1 - |\langle u_A, u_{A1}\rangle/(\|u_A\|\|u_{A1}\|)| \tag{14}$$

$$V_{A3} = \frac{1}{(n+n1)}[(n1)(x_t - \bar{x}_i)(x_t - \bar{x}_i)^T + \sum_{x_i \in A}(x_i - \bar{x}_i)(x_i - \bar{x}_i)^T] \tag{15}$$

$$V_{A3}(u_{A3}) = \lambda_{A3}(u_{A3}) \tag{16}$$

$$s = 1 - |\langle u_A, u_{A1}\rangle/(\|u_A\|\|u_{A1}\|)| \tag{17}$$

Stochastic optimization models and tradeoff analysis of evacuation problems will now be discussed. $trans_{e,f,g,h}$ is the number of people traveling from node e to f at time g for scenario h. $resident_e$ is the people that need to be evacuated at node e, and $shelter_{f,h}$ is number of people evacuated to shelter f. Additionally, $sheltercost_f$ represents the construction cost of new shelter capacity at node f. $transcost_{e,f}$ denotes the construction cost of new transportation capacity between nodes e and f. $TIMECOST_{e,f}$ is the average traveling time from node e to f. $PROB_h$ is the probability of a stochastic debris flow scenario, h. Eq. (18) calculates the weighted average of facility expansion cost, plus expected evacuation time. Eq. (19) confines the people living in the evacuation area that need to be evacuated. In Eq. (20), a mass balance equation defines whether total inflow is equal to total outflow at node e. Eq. (21) calculates the total people evacuated to shelter f for scenario h. Eq. (22) determines if all residents are evacuated. $transcap_{e,f}$ and $transnewcap_{e,f}$ are the existing and new transportation capacities between nodes e and f. The transportation capacity constraint is established in Eq. (23). The existing and new shelter capacities at node f are presented by $sheltercap_f$ and $shelternewcap_f$ and shelter capacity is formulated in Eq. (24). Nonnegativity constraints are established in Eq. (25), and according to the bag-of-words-based anomaly-detection PCA prediction, and real-time monitoring, debris flow evacuation is planned based on the stochastic optimization model in Eqs. (18)-(25).

Minimize



Natural Hazards
and Earth System
$\sum_{e,f}[\text{transcost}_{e,f} \times \text{transnewcap}_{e,f}] + \sum_f [\text{sheltercost}_f \times \text{shelternewcap}_f ]$

$\quad\quad + \sum_h [(\text{prob}_h ) \times (\sum_{e,f,g}[\text{TIMECOST}_{e,f} \times \text{trans}_{e,f,g,h}])]$ (18)

Subject to

$\sum_{f \in ADJ(e)} \text{trans}_{e,f,g,h} = \text{resident}_e$ $\quad\quad \forall\, e, f \notin m, g = 1, h$ (19)

$\sum_{f \in ADJ(e)} \text{trans}_{f,e,(g-1),h} = \sum_{f \in ADJ(e)} \text{trans}_{e,f,g,h}$ $\quad\quad \forall\, e, g \neq 1, h$ (20)

$\sum_{e \in ADJ(f),g} \text{trans}_{e,f,g,h} = \text{shelter}_{f,h}$ $\quad\quad \forall\, f, h$ (21)

$\sum_f \text{shelter}_{f,h} = \sum_e \text{resident}_e$ $\quad\quad \forall\, h$ (22)

$\text{trans}_{e,f,g,h} \leq \text{transcap}_{e,f} + \text{transnewcap}_{e,f}$ $\quad\quad \forall\, e, f, g, h$ (23)

$\text{shelter}_{f,h} \leq \text{sheltercap}_f + \text{shelternewcap}_f$ $\quad\quad \forall\, f, h$ (24)

$\text{trans}_{e,f,g,h} \geq 0, \text{shelter}_{f,h} \geq 0$ $\quad\quad \forall\, e, f, g, h$

$\text{transnewcap}_{e,f}, \text{shelternewcap}_f \geq 0$ $\quad\quad \forall\, e, f$ (25)

## 3 Results and discussion

Case studies of debris flows in Shenmu village and Fengchiu, central Taiwan, are conducted in this section. The bag-of-words and anomaly-detection PCA methods are applied to predict the occurrence of a debris flow, and the results are compared with the traditional effective accumulative rainfall prediction and red warning systems by Taiwan SWCB.

Shenmu village and Fengchiu are located in Nantou, Taiwan (Fig. 1). Shenmu village has an area of 77 km$^2$, with a population of 750 people; Fengchiu has an area of around 25 km$^2$ and a population of 770 people. Shenmu village and Fengchiu are areas of high potential debris flow risk. They are located in the Chenyulan stream watershed, and the locations are close to the convergence of the Heshe, Shenmu, and Chushui streams (Chenyulan stream's tributary). The debris flow monitoring stations and warning systems in Shenmu village and Fengchiu were established in 2002.

In Shenmu village, the Typhoon Herb in 1996 brought an accumulated rainfall of approximately 600 mm in 24 hours, and a maximum rainfall rate of 74 mm/hour. Debris flows of 450,000 m$^3$ were triggered within 5.5 hectares of Shenmu village, causing five deaths and injuring six people. Furthermore, Typhoon Toraji in 2001, Typhoon Mindulle in 2004, and Typhoon Morakot in 2009 brought a record-high rainfall that caused serious debris flows, landslide, flooding, leading to damage to life, private property, transportation, water, food, and electricity supplies in Shenmu village. Fengchiu is also an area of high

potential debris flow risk. Typhoon Herb caused two deaths and destroyed houses, fruit orchard, roads, dams, and hydraulic constructions. The Chi-Chi earthquake and Typhoon Toraji in 2001 caused severe landslides and debris flows in Fengchiu.

      A framework of debris flow detection and evacuation planning is constructed here, with the flowchart plotted in Fig. 2. Time series data for rainfall and soil water content are collected and separated into training and testing data. The bag-of-words method is applied to check the occurrence of rainfall subsequence data. After training the model, an anomaly-detection





PCA is used to measure the impact of testing data. Abnormal outlier data is detected by quantifying the change in principal
directions.

The effective accumulative rainfall method and red warnings are used by Taiwan SWCB to predict debris flows.
These two methods are compared with the bag-of-words and anomaly-detection PCA methods. Eq. (26) calculates the effective
accumulative rainfall, EAR, which is equal to the weighted sum of current rainfall $RAIN_0$ and previous rainfall $RAIN_s$. The

190 weights are one for current rainfall, and $(\alpha)^s$ for the rainfall that occurred s days previously. Taiwan SWCB assumes that
discounting factor $\alpha$ equals 0.7, and the length of time for previous rainfall data is seven days (i.e., s = 1, 2, 3, ... , 7). Taiwan
SWCB also provide a red warning for debris flow events on the basis of the EAR method, as well as meteorological data and
expert judgment.

$$EAR = RAIN_0 + \sum_s [(\alpha)^s \times RAIN_s] \qquad (26)$$

Table 1 shows the bag-of-words and anomaly-detection PCA prediction of debris flows in Shenmu village between 2011
and 2015. The bag-of-words and anomaly-detection PCA methods predict 13 anomaly data, while there are eight actual debris
flow events. This research uses two indicators to compare prediction methods, including prediction rate and prediction
accuracy (Table 1). Prediction rate measures the prediction percentage of actual debris flows, and prediction accuracy is the
correct rate of prediction. The bag-of-words and anomaly-detection PCA methods are able to predict six out of eight actual

events, and the prediction rate is 75%. The method projects 13 events, six of which are correct, and provides a prediction
accuracy of 46%.

The effective accumulative rainfall and red warning methods used by Taiwan SWCB are tested and compared with
our methods. In Table 1, both effective accumulative rainfall method and red warming predicted two out of eight actual events
between 2011 and 2015. The prediction rates are 25% for both of the Taiwan SWCB methods. The effective accumulative

rainfall method predicts eight possible events, providing a prediction accuracy of 25%. The red warning method presents a
prediction accuracy of 29% by projecting seven potential events.

The bag-of-words and anomaly-detection PCA methods are applied to predict debris flows in Fengchiu. Table 2 shows
that, between 2011 and 2015, only one actual debris flow event occurred on May 03, 2012. The bag-of-words and anomaly-
detection PCA methods are able to correctly predict the actual debris flow event, so the prediction rate is 100%. The models

detect four anomaly subsequences with one correct prediction, while a prediction accuracy of 25% is provided.

Window size, length of bag, filter ratio of training data, and anomaly thresholds are important parameters of the bag-
of-words and anomaly-detection PCA methods. Those parameters are examined for the case studies. Window size represents
the length of subsequences, and a large window size indicates greater variation of words in the bag-of-words method and a
small window size provides high computational efficiency. Length of bag is the number of words in the method. A large length

of bag provides more basic components (i.e., more words) for the bag-of-words method. The filter ratio of training data
represents the anomaly ratio of training data that should be deleted. The remaining training data are processed to train the
anomaly-detection PCA. A high filter ratio provides less normal training data, while a low ratio treats most training data as



normal. The anomaly threshold is the standard for measuring the impact of principal components and detecting anomaly data. A high anomaly threshold predicts more debris flow events with low accuracy. In contrast, a low anomaly threshold leads to high prediction accuracy, but fewer events could be simulated. Optimal parameters are determined for the case studies of Shenmu village and Fengchiu. To maximize prediction accuracy, the optimal filter ratio and anomaly threshold of Shenmu village are 0.1 and 0.025. The optimal filter ratio and anomaly threshold for Fengchiu are 0.05 and 0.02. The optimal window size and length of bag are 4 and 15. The results show that increasing window size and length of bag requires more computation, but does not provide better prediction accuracy.

In Shenmu village, the debris flows on July, 13th, 2011 and November 10th, 2011 are not predicted by our bag-of-words and anomaly-detection PCA methods. Figs. 3-6 plot the rainfall data and soil water content of those two events. Figs. 3 and 5 present 7-days of accumulative rainfall, which are 85.0 mm and 56.5 mm on the July, 13th, 2011 and November 10th, 2011 events. Figs. 4 and 6 display average soil water contents of 60.1% and 61.7% on the July, 13th, 2011 and November 10th, 2011 events. Compared with historical debris flow data, these two debris flows have relatively low accumulative rainfall and average water content. Hence, those extreme events are not predicted by the models.

## 4 Conclusions

This research establishes a bag-of-words model to analyze the occurrence of debris flows and proposes an anomaly-detection PCA method for debris flow prediction. Additionally, a stochastic programming model is formulated to optimize real-time evacuation planning. The significance of this research includes innovative development and application of machine-learning and optimization methodologies for debris flow prediction and evacuation planning.

In the case study, the bag-of-words and anomaly-detection PCA methods simulate and predict the occurrence of debris flows in Shenmu village in the Chen-Yu-Lan stream watershed, Taiwan. The results show a 75% prediction rate (of eight debris flow events) and 46% prediction accuracy (out of 13 trials). Stochastic optimization of debris flow evacuation planning is also established. Overall, a framework for debris flow detection and evacuation is developed. Optimal parameters of window size, length of bag, filter ratio of training data, and anomaly threshold are analyzed for the bag-of-words and anomaly-detection PCA methods. By optimizing these parameters, prediction accuracy of debris flows in Shenmu village and Fengchiu is increased. Future studies include improvement of debris flow prediction and detection, development of disaster warning and response systems, and practical tests and application in areas at risk of debris flows.

## Acknowledgments

The authors wish to thank the editors and anonymous referees for their thoughtful comments and suggestions. The authors are solely responsible for all opinions expressed in this paper and any remaining errors. This research was funded by the Taiwanese Ministry of Science and Technology (MOST) under Grant MOST-105-2627-M-002-037 (105T612C502).

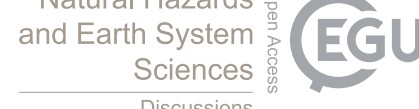



## Nomenclature

In this paper, indices and decision variables use lowercase letters. Uppercase letters indicate given coefficients. The indices,
coefficients, decision variables, and their definitions are listed below.

Indices

| | |
|---|---|
| i | time series, $i = 1, 2, \dots, n$ |
| j | centroid, $i = 1, 2, \dots, d$ |
| k | observation, $k = 1, 2, \dots, K$ |
255  s | day, $s = 1, 2, \dots, S$ |
| e, f | node of evacuation network |
| g | evacuation time |
| h | stochastic scenario |

Variables

| | |
|---|---|
A | all observed data |
| A1 | removing $x_t$ to all observed data, i.e., $A1 = A \backslash \{x_t\}$ |
| A2 | adding $x_t$ to all observed data, i.e., $A2 = A \cup \{x_t\}$ |
| A3 | adding $x_t$ to all observed data, i.e., $A3 = A \cup \{x_t, x_t, \dots, x_t\}$ |
| c | number of time series class |
265  d | number of centroids for k-means method |
| $data_k$ | data of observation K |
| EAR | effective accumulative rainfall |
| $h_j$ | index of closest centroid |
| H | index vector of $h_j$ of closest centroid |
n | number of time series |
| p | number of elements of a time series |
| $RAIN_0$ | current rainfall |
| $RAIN_s$ | previous rainfall of s days beforehand |
| U | principal directions |
$u_A$ | eigenvector of covariance matrix of data A |
| $u_{A1}$ | eigenvector of covariance matrix of data A1 |
| $u_{A2}$ | eigenvector of covariance matrix of data A2 |
| $u_{A3}$ | eigenvector of covariance matrix of data A3 |
| $V_A$ | covariance matrix of data A |





| 280 | $V_{A1}$ | covariance matrix of data A1 |
|---|---|---|
| | $V_{A2}$ | covariance matrix of data A2 |
| | $V_{A3}$ | covariance matrix of data A3 |
| | w | length of window; number of subsequence elements |
| | $x_p^i$ | element of time series vector |
| 285 | $X^i$ | time series i |
| | $y^i$ | class variables of time series $X^i$ |
| | α | discounting factor of effective accumulative rainfall method |
| | $\lambda_A$ | eigenvalue of covariance matrix of data A |
| | $\lambda_{A1}$ | eigenvalue of covariance matrix of data A1 |
| 290 | $\lambda_{A2}$ | eigenvalue of covariance matrix of data A2 |
| | $\lambda_{A3}$ | eigenvalue of covariance matrix of data A3 |
| | μ | mean of observation data |
| | ADJ(e) | adjacent nodes of node e |
| | $PROB_h$ | probability of stochastic debris flow scenario h |
| 295 | $resident_e$ | people needed to be evacuated at node e |
| | $shelter_{f,h}$ | number of persons evacuated to shelter at node f for scenario h |
| | $sheltercap_f$ | existing shelter capacity at node f |
| | $sheltercost_f$ | construction cost of new shelter capacity at node f |
| | $shelternewcap_f$ | new shelter capacity at node f |
| 300 | $TIMECOST_{e,f}$ | average traveling time cost from node e to f |
| | $trans_{e,f,g,h}$ | number of persons traveling from node e to f at time g for scenario h |
| | $transcap_{e,f}$ | existing transportation capacity between nodes e and f |
| | $transcost_{e,f}$ | construction cost of new transportation capacity between nodes e and f |
| | $transnewcap_{e,f}$ | new transportation capacity between nodes e and f |

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



**Figure captions**

**Figure 1.** Case studies of Shenmu village and Fengchiu, Taiwan.

**Figure 2.** Flowchart of bag-of-words and anomaly-detection PCA debris flow prediction algorithms and stochastic programming of evacuation planning.

**Figure 3.** Rainfall of debris flow events in July, 13th, 2011.

**Figure 4.** Soil water content of debris flow events in July, 13th, 2011.

**Figure 5.** Rainfall of debris flow events in November 10th, 2011.

**Figure 6.** Soil water content of debris flow events in November 10th, 2011.





**Table 1.** Bag-of-words and anomaly-detection PCA predictions of debris flows on Shenmu village.

| | Actual debris flow events | Prediction method | | |
| --- | --- | --- | --- | --- |
| | | Bag-of-words and anomaly-detection PCA | Effective accumulative rainfall method | Red warning |
| Event time (UTC +8:00 Taipei time) | 2011/07/13/14:30 | 2011/07/19/03:30 | 2012/06/10/20:30 | 2012/06/10/17:00 |
| | 2011/07/19/23:00 | 2012/05/03/17:40 | 2013/05/21/05:40 | 2012/08/02/14:00 |
| | 2011/11/10/03:20 | 2012/05/20/09:30 | 2013/07/13/06:50 | 2013/07/13/03:00 |
| | 2012/05/04/16:00 | 2012/06/10/12:10 | 2013/08/22/05:50 | 2013/08/21/20:00 |
| | 2012/06/10/19:00 | 2012/08/02/06:20 | 2014/07/23/16:30 | 2014/07/23/14:00 |
| | 2013/05/19/07:00 | 2012/08/16/21:50 | 2015/05/25/02:10 | 2015/08/08/17:00 |
| | 2013/07/13/07:00 | 2013/05/17/05:50 | 2015/08/09/02:10 | 2015/09/29/02:00 |
| | 2014/05/20/12:50 | 2013/05/19/08:40 | 2015/09/29/03:20 | |
| | | 2013/05/21/06:30 | | |
| | | 2013/07/13/02:00 | | |
| | | 2013/08/21/21:10 | | |
| | | 2014/05/20/14:10 | | |
| | | 2015/05/26/15:10 | | |
| Prediction rate[a] | | 75% | 25% | 25% |
| Prediction accuracy[b] | | 46% | 25% | 29% |

[a] Prediction rate: prediction percentage of actual debris flow.

[b] Prediction accuracy: rate of accurate prediction.



**Table 2.** Bag-of-words and anomaly-detection PCA prediction of debris flows in Fengchiu.

| | Actual debris flow events | Prediction method | | |
| --- | --- | --- | --- | --- |
| | | Bag-of-words and anomaly-detection PCA | Effective accumulative rainfall method | Red warning |
| Event time (UTC +8:00 Taipei time) | 2012/05/03/16:00 | 2012/05/03/15:40 2012/05/20/09:50 2012/08/02/06:50 2013/05/19/09:20 | 2012/06/12/13:40 2012/08/02/08:50 2013/07/13/07:00 2013/08/22/07:20 | 2012/06/11/20:00 2012/08/02/08:00 2013/07/12/23:00 2013/08/21/20:00 |
| Prediction rate[a] | | 100% | 0% | 0% |
| Prediction accuracy[b] | | 25% | 0% | 0% |

[a] Prediction rate: prediction percentage of actual debris flow.

[b] Prediction accuracy: rate of accurate prediction.





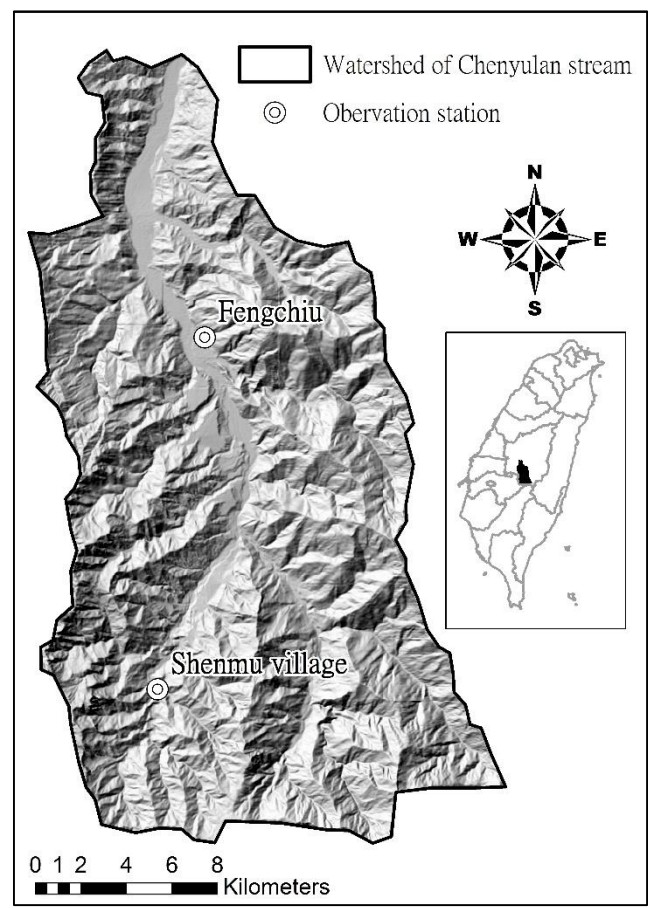

**Figure 1.** Case studies of Shenmu village and Fengchiu, Taiwan.





Bag-of-words

Subsequences with window $w$ are extracted from time series data

k-means method is applied to cluster subsequences in $d$ centroids

Testing time series data are categorized by occurrence of centroids

Time series data of rainfall and water content

Training
Testing

Anomaly-detection PCA

Calculates covariance matrix $V$

Computes principal components by maximizing variance or minimizing reconstruction error

The impact of data on principal components is measured

Debris flow prediction and warming

Debris flow evacuation using stochastic optimization models

**Figure 2.** Flowchart of bag-of-words and anomaly-detection PCA debris flow prediction algorithms and stochastic programming of evacuation planning.





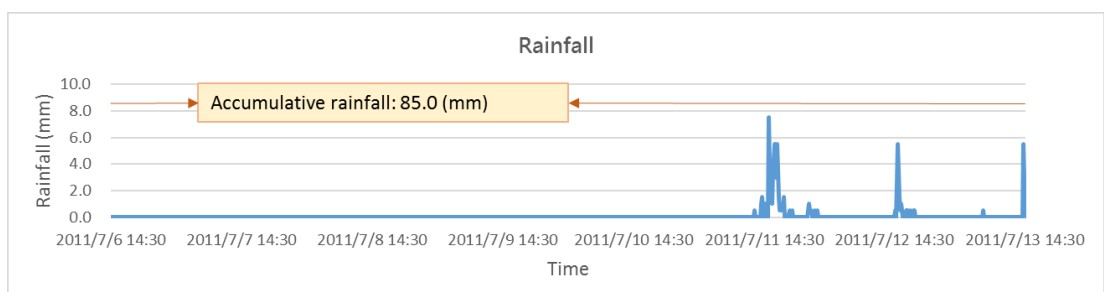

**Figure 3.** Rainfall of debris flow events in July, 13th, 2011.



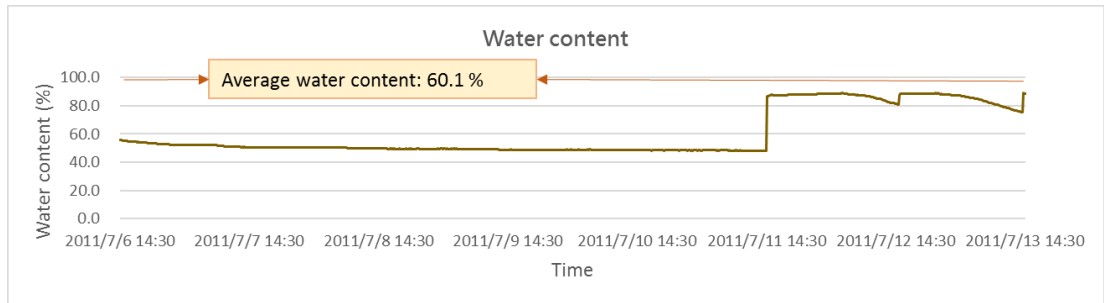

**Figure 4.** Soil water content of debris flow events in July, 13th, 2011.




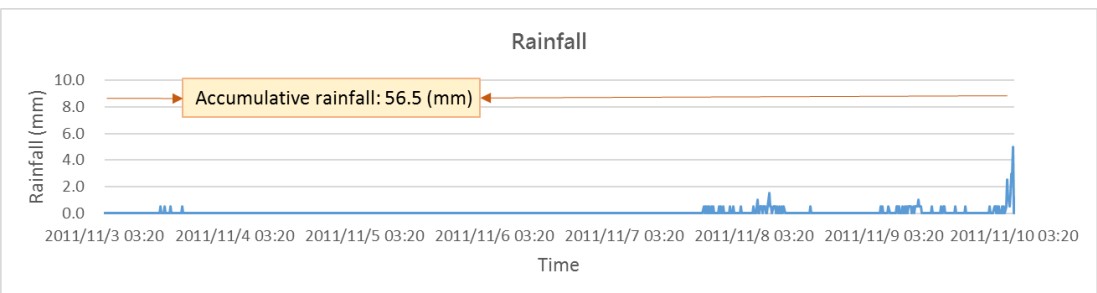

**Figure 5.** Rainfall of debris flow events in November 10th, 2011.



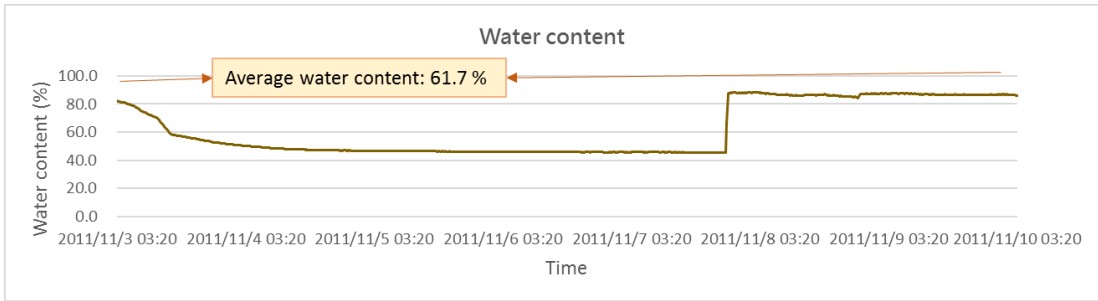

**Figure 6.** Soil water content of debris flow events in November 10th, 2011.