# Peer review of "Bag-of-words-based anomaly-detection principal component analysis and stochastic optimization for debris flow detection and evacuation planning"

_Natural Hazards and Earth System Sciences, 2017_

## Referee Comment (RC1) · Anonymous Referee #1 · 15 Nov 2017

The ms presents an interesting model to detect debris flows by a "bag-of-words" based procedure and to improve the evacuation planning by stochastic optimization technique. The model is applied to two villages located in Taiwan.

The topic is adequate for NHESS and the proposed model is rather novel, but unfortunately, it cannot be accepted in the present form, since there are many major and minor critiques that should be corrected before acceptance (or re-submission).

Major critiques:

[Figure]

1) The explanation of the model is not very clear and really difficult to follow for a reader, who is not expert in machine-learning and stochastic analysis. I propose two major improvements: a) the structure of the entire model and its different parts must be clearly described at the beginning of the Method-Section. You should use Fig. 2, which is now in the Result-Section, for this general description. b) You may use less equations and more explanations. The description is now very complex: you incorporate 25 equations and many (too much) parameters.

2) The Result-section must significantly improve, because there is a lack of information or description. In the present form (now only two pages!), two tables are the main results. The four graphs in Fig 3-6 are not clear and only commented in 6 lines without adding details, interpretation etc. The lack of explanations and details is a major problem throughout this section. Some examples: a) Time series for rainfall and soil water content are analysed without basic information on the location of these sensors, scan rate, data transmission etc. b) No values and details on the thresholds are given. In L185, the authors introduce a rainfall method and warning without additional explanations or at least references. c) Different parameters were examined (L211-212), but no values (or thresholds) of this sensibility analysis is discussed in detail. I propose to show some graphs on this topic.

3) The authors mention a separation into training and test data (L183), but after reading the Result-section, it's not clear, what was training and what was test/validation of the model. Please, clarify this aspect.

4) The part of the evacuation planning is explained in the Method-section, but not even mentioned in the Results. Please comment this.

5) Introduction-section must be improved: - L34-51 is a long description of the methodology and must principally be placed in the Methods-Section. - the description of the goals is missing

6) Add information on the monitoring stations (already criticised in 2a). Location and

detailed description of all sensors are necessary to understand the results. Only some doubts that I have: how the exact time of debris flow was determined? The detection of the events is recorded near the villages, not in the debris flow initiation zone (normally there is a delay between triggering and arrival in the village).

Minor critiques: I) English should be corrected and improved. II) citation must be corrected (L25: order; L109: format etc.) III) The Section "Results and discussion" lacks of a substantial discussion. IV) L175-181: improve the description of the debris flow episodes and add references. "Debris flows of 450000m3 were triggered" is not very illustrative. V) L199: explain, how you define a "correct prediction" (prediction within the same day, the same floating 24h-window etc.?) VI) "remaining training data" is not clear VII) L225-230: as explained above, additional information and discussion is needed (soil water content sensor is at which depth? How many sensors do you use? What is the "average water content" and how it was determined? etc.) VIII) Table 1 and 2: indicate the correct predictions by bold event times IX) Figures must be improved: - more figures are necessary. - Fig 3-6 are unsubstantial in the present form and must be improved. Merge Fig3-4 and Fig5-6. Add arrows to indicate debris flow triggering, warning levels etc.

---

## Referee Comment (RC2) · Anonymous Referee #2 · 6 Dec 2017

**1   General Statemets**

Before we get to the guts of the article I would like to thank the editor for the oppurtunity to review this article. I think the idea of the article is really nice and it was a interessting journey to review this paper. First I present a short summary after which the most prominent results of the review are stated. Then an sometimes exhaustive description of the points stated before are given. I hope the review is helpfull and not offending in any way. Lets get dirty!

[Figure]

**2   Summary**

The authors use a bag-of-words transformation of rainfall and soil water content time series to classify, with a classifier in form of an anomaly detection based on oversampling PCA, into debris flow and non debris flow measurement series. They compare the accuracy of the classifier with two best practice methods for two case studies. Also a stochastic evacuation model is presented which should use the results of the classification model as an imput; but no further eloboration nor usage of the model is made.

2.1   The Review Result in 2 Sentence and a Short List

The classification algorithm, as well as the evacuation model serve as a solid basis for a publication in this very journal. Alltough a major overhaul of the article is needed including

- Major improvements regarding the method chapter in a more detailed less equation droping way. The algorithms scream for visual descriptions!

- Major impovement of the results by making a more in depth comparison of the new classifier (more data if possible) and also using the proposed evacuation model

- Improvement of the Introduction: focus on time series classification in the context of natural hazards

**3   Critique on the Introduction**

**3.1   Critique 1**

The Introduction is not on the point

**3.1.1   Comments and Possible Solution on Critique 1**

I think the introduction should be mainly concerned with the stae of the art of debris flow forecast based on different measurement series like rainfall, seismic signals, infrasound and at which state the are in the sense of warning times and accuracy. Then the auhtors should present the usage of the bag-of-words as well as the anomaly dectection PCA in the context of time series especially in the case of natural hazards.

**4   Critique on the Methods**

In this section I describe the problems with the methodology chapter

**4.1   Critique 1**

The bag-of-words model is poorly described.

**4.1.1   Comments on Critique 1**

The authors try to explain the classification algorithm by first introducing the structure of the data, but in the same moment they also give specifications of the algorithm. A few

seconds later the authors return to the structure of the data and suddenly information about the data splitting is given, only to return a second later to the algorithm itself. Alltough I am a big fan of chaotic structure when it comes to music; in this case it is not charming at all. To be honest it makes it somehow diffcult to understand and I don't know if I understand how the authors build their model.

In my humble opinion I think the authors tried to state the following: the problem starts with $n$ measurement series of derbis flow related variables. Because the measurements are functions of time they are called *time series*. The length of a time series (or the number of measurements per time series) is $p$. Also known, for every time serires is, from which process it is originating, so that every time series has a class label. In the case of the article the class is *debris flow* or *no debris flow*. At this point in time the data set should have the, or similar structure, as shown in the table below.

|  | time series |  |  |  | debris flow |
| --- | --- | --- | --- | --- | --- |
| $x_{1,1}$ | $x_{1,2}$ | $x_{1,3}$ | $\ldots$ | $x_{1,p}$ | yes |
| $x_{2,1}$ | $x_{2,2}$ | $x_{2,3}$ | $\ldots$ | $x_{2,p}$ | no |
| $x_{3,1}$ | $x_{3,2}$ | $x_{3,3}$ | $\ldots$ | $x_{3,p}$ | yes |
| $\vdots$ | $\vdots$ | $\vdots$ | $\vdots$ | $\vdots$ | $\vdots$ |
| $x_{i,1}$ | $x_{i,2}$ | $x_{i,3}$ | $\ldots$ | $x_{i,p}$ | yes |
| $\vdots$ | $\vdots$ | $\vdots$ | $\vdots$ | $\vdots$ | $\vdots$ |
| $x_{n,1}$ | $x_{n,2}$ | $x_{n,3}$ | $\ldots$ | $x_{n,p}$ | yes |

This is the basis for constructing and testing the classifier and finally using it to classify time series were the class is unknown. Following the bag-of-words model one has to construct a collection of words, which is often refered to as a codebook Reference 1 or dictionary Reference 2. This gives raise to the problem of defining words in the context of a time series. As I understood it the authors did the follwoing: first a window of size $w$ is slided along the time series with step size $1$. Below is a toy example, with $w = 4$

illustrating this.

| $x_1$ | $x_2$ | $x_3$ | $x_4$ | $x_5$ | $x_6$ | $x_7$ | $x_8$ | $x_9$ | $x_{10}$ | window |
|---|---|---|---|---|---|---|---|---|---|---|
| 10 | 15 | 20 | 25 | 30 | 31 | 40 | 45 | 60 | 25 | $w_1$ |
| 10 | 15 | 20 | 25 | 30 | 31 | 40 | 45 | 60 | 25 | $w_2$ |
| $\vdots$ | $\vdots$ | $\vdots$ | $\vdots$ | $\vdots$ | $\vdots$ | $\vdots$ | $\vdots$ | $\vdots$ | $\vdots$ | |
| 10 | 15 | 20 | 25 | 30 | 31 | 40 | 45 | 60 | 25 | $w_6$ |

This results in a collection of $10 - (4 - 1) = 6 = p - (w - 1)$ possible words defined by $4 = w$ characters as shown in the next table (a lot of tables here, I am sorry for that).

| $c_1$ | $c_2$ | $c_3$ | $c_4$ |
|---|---|---|---|
| 10 | 15 | 20 | 25 |
| 15 | 20 | 25 | 30 |
| $\vdots$ | $\vdots$ | $\vdots$ | $\vdots$ |
| 40 | 45 | 60 | 25 |

This collection of $w$-long words is repeated for all available observations so that the above table is expanded to $n * 6 = n * (p - (w - 1))$ entries. The authors use k-means clustering to generate distinct groups of words which is according to Reference 1 and 2 among others a standard approach to build the dictionary which I understand as the analogon of linguistic processing of words to groups like *typical spam mail word*, *non typical spam mail word* etc. After this procedure the authors should be able to build a histogram of the counts of words from their codebook which should serve as the basis for their classification model.

**4.1.2 Possible Solution to Critique 1**

Have a look a the description of the bag-of-words model by Reference 1 which is nice and clear and use it as a guide to explain it for your article. There is also nothing dirty to it if you describe it in a more direct way for your data droping some gernality in the equations (notationwise) will hurt no one because the paper is an accplication non theoretical driven one.

**4.2 Critique 2**

It is not clear how the test series are classified in other words which classifier is used for the established bag-of-word represantation?

**4.2.1 Comments on Critique 2**

To come to this point I had to get some knowledgde about the bag-of-words model and especially how it can be used to classify time series. I hope that I got everything wright to this point. So what we have are histograms of words (or the frequency of clusters for the subsequences per series) for known classes. The next step is to build a classifier based on those classes which seems to be a simple nearest neighbor classifier.

**4.2.2 Possible Solution to Critique 2**

State more clearly the workflow of the bag of words model. Critique 2 should vanish if you change the structure according to critique 1.

**4.3 Critique 3**

How does the anomaly-detection PCA fit into the modeling framework, in other words for what is it used?

**4.3.1 Comments on Critique 3**

I hope this is not a stupid critique but it is not clear to me what purpose the PCA is serving.[1] After studing Reference 3 I could follow the presentation of the authors but the explanation in this article is poor and and a lot of the given equations are unnecessary. Let me elobrate on this: First we have the descriptions of the PCA which is formulated as the solution of two different optimizations problems (equation 5 and 6). I don't think that this is necessary because PCA is a well known method and a simple statemend of solving the eigenvalue problem would be sufficient especially in the context that the eigenvalue problem is also used in the anaomly detection formulation. Also the introduction of more variables like: $\text{data}_k$ and $K$ is adding more shadows than lights to this chapter. Especially because $k$ is the standard variable for indicating the number of cluster centers. After equation 6 the authors get to the core of the anomaly dectection based on PCA. The idea, regarding (), is to add or leave out an observation and see how the first, or most important, principal component is changing its direction in comparision to the first principal component based on the whole data set. This idea can also be used in an online sense, meaning that at the time new data arrives the algorithm is used to check wether or not the new observation is an outlier or a normal observation.[2] The equation 7 to 17 are descriping this in a mathematical sense:
* * *
[1]Is it the classifier for the bag of words? So that the data entering the PCA is the histogram $H$?

[2]please note that I use *normal* and *outlier* in a statistical sense which means their is some region of acceptable probability of having an observation with that value which would translate into *normal* and outside would therefore translate into *outlier*. In the case of this algorithm the normal region is definied by the magnitude 1 minus of the absolute cosine similarity
Equation 7 says comupte the covariance of the whole data set $A$, which is also given in equation 4.[3] The exact same calculation is essentially given in equation 8, 12 and 15 which is the updating of the covariance matrix based on leaving out one sample, adding one sample or adding n times the same sample. Also the equations have notational problems: what $i$ is the $\bar{x}_i$ in equation 12 in the left part of the sum? The repetition of equations similar to 7, 8, 12 and 15 is true for 9, 10, 13, 16 which states the eigenvalue problem depending on the covariance matrices given in 8, 12, and 15. Equations 11, 14 and 17 are essentially cosine similarities based on the first principal component eigenvector depenting on the leave one out, add one or add n instances to the data and are the variable which defines which observation is or is not an outlier.

**4.3.2 Possible Solution to Critique 3**

A possible solution maybe twofold

1. state clearly the purpose of the anomaly detection by oversampling PCA

2. describe the method in three ways first in an general sense by words, second by a code listing and third by stating the key equations

All this statements should be connected to your problem especially how the PCA is used as a classifier for the bag-of-words representation of your time series.

**4.4 Critique 4**

The stochastic optimization model is poorly described and not even used in the article
* * *
[3]the auhtors follow the notation given in () but not their own given in equation 4 to 6. If I got it right the equivalence should be $K = A$ and $\text{data}_k = x_i$

**4.4.1  Comments on Critique 4**

Okay their seems to be some pattern: the auhtors use very interessting approaches but the delivery of the methods is poor. To understand the stochastis optimisation the reader has to fill a lot of gaps. First of all the notation which is hard to follow especially because some of the indices are not even described. Again I will try to retell what I think the authors try to say: there are sources and there are sinks which the auhtors call $e$ and $f$. I assume that their are more then one shelter and source, which means that $e$ and $f$ are vectors $\mathbf{e}$ and $\mathbf{f}$.[4] For now I assume just one scenario so that I can drop the index $h$. This said I try to restate the optimization problem proposed by the authors. First the constrains in equation 25 mean, that their cannot be a negative number of people in a shelter (authors: $\mathrm{shelter}_f \geq 0$). That the number of people transported from any source to any shelter is not negative (authors: $\mathrm{trans}_{e,f,g} \geq 0$), which means that their are no people traveling back from a shelter. And finally two constrains are given that the existing capacity cannot be reduced (authors: $\mathrm{transnewcap}_{e,f} \geq 0$ and $\mathrm{shelternewcap}_f \geq 0$).

The constrains in equations 23 and 24 regard the number of people in shelter and number of people currently beeing transported which can not be higher than the maximum shelter and trasnport capacity at time $g$. The constrain formulated in euqation 19 is missleading: the number of residents that need to be evacuated in a sourece $e_i$ is equal to the sum over all people that are transported from $e_i$ to a subset of $\mathbf{f}$ at the time instance $1$. That means all people that should be evacuated should be evacuated. I think the autors wnated to state that the total number of people per source or in other words: first determinate the sources under scenario $h$, then look at to which shelter every source is connected and count the number of people which must fit the shelter. The confusion goes on when constrain 20 is regarded whcih states the number of people leaving the source $e_i$ at time $g-1$ must be equal the number transported at time $g$.
* * *
[4]because equation 18 is summing over all $e$ and $f$, I assume this is true

So what the authors set as constrain is that what is leaving $e_i$ is euqal what is leaving $e_i$. The constrain 21 is not a constrain just the number of people at shelter $f_i$ at time $g$ which is the number of people in shelter $f_i$ at time $g-1$ plus the number of people entering $f_i$ at time $g$. Constrain 22 is more a stopping criteria than a constrain which states that the simulation is done when all people are transported from the sources to the shelter. Finally I can have a look at the cost function which should be minimized. The first term are the costs for new transport capacity, the second term are the costs for new shelter capacity and the third is the probability weighted costs for the number of people that must be evacuated per scenario depending on time needed do evacuate all of them.

Overall the model is confusing and I cannot see the connection between the bag-of-words PCA classifier and the evacuation model. I tried to get the idea behind the model and build it in a spreadsheet programm assuming a simple set up with three sources and three shelter calculating one scenario. Maybe the authors better understand my concerns when the can have a look at how I understand their model in action.

4.4.2   Possible Solution to Critique 4

First of try to state the model in a more clear description by changing the variables in something like given in the follwoing table.

[Figure]

| Variable | Description | proposed Symbol |
|---|---|---|
| shelter$_f$ | number of people in shelter $f$ at time $g$ | $v_j(t)$ |
| sheltercap$_f$ | maximum capacity of shelter $f$ at time $g$ in number of people | $V_j(t)$ |
| shelternewcap$_f$ | increase of shelter capacity of shelter $f$ at time $g$ in number of people | $\Delta V_j(t)$ |
| trans$_{e,f,g}$ | number of people traveling from $e$ to $f$ at time $g$ in number of people | $q_{i,j}(t)$ |
| transcap$_{e,f}$ | maximum transport capacity of people from $e$ to $f$ at time $g$ in number of people | $Q_{i,j}(t)$ |
| transrnewcap$_{e,f}$ | increase of transport capacity of route $e$ to $f$ at time $g$ in number of people | $\Delta Q_{i,j}(t)$ |
| sheltercost$_f$ | costs of building new shelter capacity for shelter $f$ in monetary units per person | $C_{V_j}$ |
| transcost$_{e,f}$ | costs of increasing the transport capacity from $e$ to $f$ in monetary units per person | $C_{Q_{i,j}}$ |
| TIMECOST$_{e,f}$ | costs of evacuating people from $e$ to $f$ in monetary units per person | $C_{q_{i,j}}$ |

This represantation can be further improved by droping the time and just state that this is a model which is depending on a fixed time step. After the desciption is made more clearer the model should also be used in the case study especially by using the two classification methods which I think give the secnarios $h$. Some results especially for some extrem points should be given which conclusion can be drawn from the model?

Is the transport capacity or the shelter size the most important variable? At which stochastic scenario is the importance changing if it is changing at all? What are the conclusion for evacuation planing based on the model results?

**5   Critique on the Results and Conclusions**

As stated at the beginning and also trough out the discussion about the method chapter the results and conclusion are very short. Also their presentation is poor (I know I sound like a donkey always saying the same thing). From critique to solutions: do enhance the results add na diagramm which shows a time series of a debris flow and show at which time the warnigs from which method are given also show how those methods see the time series the rainfall method only see a combination of decaying rainfall sums, while the bow-PCA methods sees data points in the PCA space trying to dected anomalies. Also show time series were erros were made because this is also interesseting. Also be more precise about the calibration or training of your model state it in a more compact way maybe a table showing the configuration and diagramms showing how the accuracy values are chaning.

**References**

**Title:** Bag-of-words Representation for Biomedical Time Series Classification
   **DOI:** 10.1016/j.bspc.2013.06.004
**Title:** The great time series classification bake off: a review and experimental evaluation of recent algorithmic advances
   **DOI:** 10.1007/s10618-016-0483-9
**Title:** Anomaly Detection via Online Oversampling Principal Component Analysis
   **DOI:** 10.1109/TKDE.2012.99

Please also note the supplement to this comment:
https://www.nat-hazards-earth-syst-sci-discuss.net/nhess-2017-325/nhess-2017-325-RC2-supplement.zip

Interactive
comment